# A Postsynaptic Density Immediate Early Gene-Based Connectome Analysis of Acute NMDAR Blockade and Reversal Effect of Antipsychotic Administration

**DOI:** 10.3390/ijms24054372

**Published:** 2023-02-22

**Authors:** Annarita Barone, Giuseppe De Simone, Mariateresa Ciccarelli, Elisabetta Filomena Buonaguro, Carmine Tomasetti, Anna Eramo, Licia Vellucci, Andrea de Bartolomeis

**Affiliations:** 1Laboratory of Translational and Molecular Psychiatry, Unit of Treatment-Resistant Psychosis, Section of Psychiatry, Department of Neuroscience, Reproductive Sciences and Odontostomatology, University Medical School of Naples “Federico II”, 80131 Naples, Italy; 2Lundbeck LLC, Deerfield, IL 60015, USA

**Keywords:** *Homer1a*, brain network, asenapine, antipsychotics, connectomics, functional connectivity, postsynaptic density, ketamine, psychosis, schizophrenia

## Abstract

Although antipsychotics’ mechanisms of action have been thoroughly investigated, they have not been fully elucidated at the network level. We tested the hypothesis that acute pre-treatment with ketamine (KET) and administration of asenapine (ASE) would modulate the functional connectivity of brain areas relevant to the pathophysiology of schizophrenia, based on transcript levels of *Homer1a*, an immediate early gene encoding a key molecule of the dendritic spine. Sprague–Dawley rats (*n* = 20) were assigned to KET (30 mg/kg) or vehicle (VEH). Each pre-treatment group (*n* = 10) was randomly split into two arms, receiving ASE (0.3 mg/kg), or VEH. *Homer1a* mRNA levels were evaluated by in situ hybridization in 33 regions of interest (ROIs). We computed all possible pairwise Pearson correlations and generated a network for each treatment group. Acute KET challenge was associated with negative correlations between the medial portion of cingulate cortex/indusium griseum and other ROIs, not detectable in other treatment groups. KET/ASE group showed significantly higher inter-correlations between medial cingulate cortex/indusium griseum and lateral putamen, the upper lip of the primary somatosensory cortex, septal area nuclei, and claustrum, in comparison to the KET/VEH network. ASE exposure was associated with changes in subcortical-cortical connectivity and an increase in centrality measures of the cingulate cortex and lateral septal nuclei. In conclusion, ASE was found to finely regulate brain connectivity by modelling the synaptic architecture and restoring a functional pattern of interregional co-activation.

## 1. Introduction

Administration of N-methyl-D-aspartate receptor (NMDAR) non-competitive antagonists has been considered as a proxy model for psychosis, characterized by NMDAR hypofunction, a molecular hallmark associated with the pathophysiology of schizophrenia, which affects several patterns of functional connectivity between brain regions [1], thus is suitable for testing the differential effects of antipsychotics.

GWAS analyses [2], post-mortem studies [3], and preclinical models [4] of psychosis and schizophrenia have highlighted significant alterations in post-synaptic density, an electron-dense region localized at the postsynaptic sites of glutamatergic synapses, with selective involvement of smaller dendritic spines, which are strongly related to learning and behavioral flexibility, resulting in reduced plasticity of brain circuits [5,6,7]. For instance, a dysregulation in protein levels of several molecular components of dendritic spines, including postsynaptic density 95 (PSD-95), NMDAR subunit GluN1, spinophilin, and Homer 1, has been detected in multiple brain regions of patients affected by schizophrenia [3,6,8,9].

Antipsychotics have been found to modulate synaptic plasticity and metaplasticity, as well as affect postsynaptic sites. Nonetheless, if the ability of antipsychotics to induce IEGs and tune molecular processes involved in synaptic regulation, may finally hesitate in the restoration of functional connectivity, is yet to be clarified [10,11]. In a previous study, we generated a functional brain network by mapping the expression of *Homer1a* evoked by the acute administration of the typical antipsychotic haloperidol and investigated differences in discrete brain network properties, as compared to the vehicle (VEH) [12].

The present work was conceived to investigate the effects of an atypical antipsychotic, asenapine (ASE), both at the level of gene expression and functional connectivity in a pharmacological model of acute psychosis. Among the characteristics of its receptor profile, beyond D2 receptor (D2R) antagonism, asenapine shows a significant antagonism with relevant affinity at D1 receptors (D1Rs), compared to other antipsychotics [13]. The action at D1R sites is even more attractive since D1R and NMDAR couple at the postsynaptic site in dendritic spines, enhancing a reciprocal activity through a positive feedback mechanism [14]. Therefore, we aimed at exploring the ASE-induced changes in *Homer1a* expression, a direct marker of synaptic activity, in 33 regions of interest (ROIs) (listed in Table 1) within the cortex, the caudate-putamen, and the nucleus accumbens of rats previously challenged with acute ketamine (KET) administration by quantitative in situ hybridization histochemistry (ISHH). We opted to study the action of these compounds on cortical and striatal structures. Indeed, cortical connectivity appears to be impaired in schizophrenia, along with the morphological finding of grey matter volume loss [15] both in the disease and during the course of treatment. Furthermore, as striatal dopamine function has been related to the pathophysiology of schizophrenia [16], and striatal structures are known to be involved in the action of antipsychotic drugs, we also investigated gene expression and connectivity of these subcortical regions.

A graph theoretical analysis and a statistical comparison between networks were applied to define group-individual differences within functional connectivity profiles.

## 2. Results

### 2.1. Gene Expression Analysis: Comparison between Experimental Groups in Homer1a mRNA Levels

*Homer1a* gene expression was analyzed in 33 subcortical and cortical regions of the rat brain by quantitative ISHH and the resulting values were compared between groups of treatment.

The dependent variable was normally distributed for each combination of the levels of the between- and within-subject factors, as revealed by a Shapiro–Wilk test, which did not give significant results. Mauchly’s test of sphericity indicated that the assumption of sphericity was violated, then we applied the Huynh–Feldt correction for degrees of freedom. There was a statistically significant two-way interaction between treatment and ROI, F(21.56, 115) = 2.096, *p* = 0.002, partial η2 = 0.308.

The Student’s *t* test was used to compare the transcript values of *Homer1a* in each ROI between (i) VEH/VEH vs. KET/VEH groups to highlight the differences in *Homer1a* expression between a psychosis-like model and normal conditions (ii) VEH/VEH vs. VEH/ASE groups, to assess the impact of the antipsychotic on the gene expression under baseline conditions; (iii) KET/VEH vs. KET/ASE groups to evaluate the effects of the antipsychotic in an animal model of psychosis. The results from the comparisons between groups are displayed in Figure 1.

The comparisons of *Homer1a* mRNA levels between VEH/VEH and KET/VEH groups are detailed in Table 2. With regard to the cortical regions, *Homer1a* expression was significantly lower in KET/VEH group compared to VEH/VEH group in the cingulate cortex (cg2 and cg1) (95% CI, −0.47 to −0.09, *t*(8) = −3.48, *p* = 0.008; 95% CI, −0.64 to −0.20, *t*(8) = −4.43, *p* = 0.002, respectively), in primary and supplementary motor cortex (95% CI, −0.51 to −0.10, *t*(8) = −3.47, *p* = 0.008; 95% CI, −0.59 to −0.12, *t*(8) = −3.56, *p* = 0.007, respectively), in the forelimb, jaw region, and dysgranular zone of the somatosensory cortex (CI, −0.44 to −0.07; *t*(8) = −3.22, *p* = 0.012; CI, −0.38 to −0.02, *t*(8) = −2.59, *p* = 0.032; CI, −0.37 to 0.00, *t*(8) = −2.28, *p* = 0.05, respectively). *Homer1a* expression was found reduced in the KET/VEH group compared to the VEH/VEH group even in subcortical areas. Specifically, in the KET/VEH group, reduced *Homer1a* mRNA levels were detected in all striatal subregions (in CPDM, CI, −0.55 to −0.05, *t*(4.96) = −3.14; *p* = 0.026; CPDL, CI, −0.57 to −0.11, *t*(5.81) = −3.63; *p* = 0.012; CPVL, CI, −0.57 to −0.03, *t*(4.65) = −2.93, *p* = 0.036; CPVM, CI, −0.69 to −0.15, *t*(4.72) = −4.09, *p* = 0.011), in the nucleus accumbens (the core, CI, −0.51 to −0.00, *t*(4.43) = −2.73, *p* = 0.047, the shell CI, −0.58 to 0.00, *t*(5.32) = −2.44, *p* = 0.05), the ventral region of the lateral septal nuclei (CI, −0.27 to −0.01, *t*(8) = −2.59, *p* = 0.033), the Calleja’s islands (CI, −0.24 to −0.05, *t*(8) = −3.55, *p* = 0.007), the ventral pallidum (CI, −0.32 to 0.10, *t*(8) = −4.21, *p* = 0.003), and olfactory tubercle (CI, −0.36 *t* −0.03, *t*(8) = −2.67, *p* = 0.028) (Figure 1). Nonetheless, significant values did not survive after the Bonferroni correction.

VEH/VEH vs. VEH/ASE comparisons are shown in Table 3. Even though almost all of the values were not significant, ASE administration resulted in higher *Homer1a* transcript levels in cg2 (CI, −0.36 to −0.01, *t*(8) = −2.43, *p* = 0.041) and lower levels in the ventral pallidum (CI, −0.023 to −0.02, *t*(8) = −2.67, *p* = 0.028)(Figure 1). However, significant values did not survive after the Bonferroni correction.

Noteworthy, the administration of ASE in an animal model of acute psychosis mimicked by acute KET exposure was able to restore *Homer1a* expression almost in all the regions considered, as outlined in Table 4. In cortical regions, the KET/ASE group exhibited higher significant levels of *Homer1a* mRNA compared to KET/VEH in the cingulate cortex (Cg2 and Cg1) (95% CI, −0.40 to −0.05, *t*(8) = −3.00, *p* = 0.017; 95% CI, −0.49 to −0.03, *t*(8) = −2.59, *p* = 0.032, respectively), in the supplementary and primary motor cortex (95% CI, −0.52 to −0.06, *t*(8) = −2.89, *p* = 0.020; 95% CI, −0.51 to −0.03, *t*(8) = −2.60, *p* = 0.032, respectively), in the forelimb region (95% CI, −0.55 to −0.09, *t*(8) = −3.19, *p* = 0.013), jaw region (95% CI, −0.53 to −0.00, *t*(8) = −2.32, *p* = 0.049), and dysgranular zone of the primary somatosensory cortex (95% CI, −0.53 to −0.02, *t*(8) = −2.50, *p* = 0.037), in the granular and dysgranular insular cortex (95% CI, −0.61 to −0.03, *t*(8) = −2.52, *p* = 0.036; 95% CI, −0.66 to −0.11, *t*(8) = −3.21, *p* = 0.012, respectively), dorsal and ventral agranular insular area (95% CI, −0.63 to −0.12, *t*(8) = −3.42, *p* = 0.009; 95% CI, −0.54 to −0.05, *t*(8) = −2.83, *p* = 0.022, respectively), in the claustrum (95% CI, −0.55 to −0.07, *t*(8) = −2.94, *p* = 0.019), and in the piriform cortex (95% CI, −0.50 to −0.03, *t*(8) = −2.66, *p* = 0.029), and in the dorsal endopiriform nucleus (95% CI, −0.36 to −0.04, *t*(8) = −2.91, *p* = 0.02). Following KET pre-treatment, several subcortical regions showed a higher *Homer1a* gene expression after ASE administration in comparison to VEH, including the lateral stripe of striatum (95% CI, −0.58 to −0.10, *t*(8) = −3.28, *p* = 0.011), the dorsomedial, dorsolateral, ventrolateral, and ventromedial caudate-putamen (95% CI, −0.52 to −0.21, *t*(8) = −5.41, *p* = 0.001; 95% CI, −0.7 to −0.32, *t*(8) = −6.24, *p* < 0.001; 95% CI, −0.84 to −0.42, *t*(8) = −6.93, *p* < 0.001; 95% CI, −0.71 to −0.21, *t*(8) = −4.32, *p* = 0.003, respectively), the core and shell of nucleus accumbens (95% CI, −0.58 to −0.11, *t*(8) = −3.36, *p* = 0.01; 95% CI, −0.63 to −0.13, *t*(8) = −3.49, *p* = 0.008, respectively), the dorsal, intermediate, and ventral septal nuclei (95% CI, −0.46 to −0.07, *t*(8) = −3.13, *p* < 0.014; 95% CI, −0.26 to −0.03, *t*(8) = −2.90, *p* < 0.02; 95% CI, −0.38 to −0.06, *t*(8) = −3.14, *p* = 0.014, respectively), the septohippocampal nucleus (95% CI, −0.36 to −0.08, *t*(8) = −3.58, *p* = 0.007), the medial septum (95% CI, −0.29 to −0.1, *t*(8) = −4.84, *p* = 0.001), the Calleja islands (95% CI, −0.47 to −0.04, *t*(8) = −2.71, *p* = 0.027), and the ventral pallidum (95% CI, −0.48 to −0.04, *t*(8) = −2.70, *p* = 0.027). No differences between groups were found in the *Homer1a* transcript values in the indusium griseum, the oral surface of the jaw region, and upper lip of the somatosensory cortex, the nucleus of the vertical limb of the diagonal band, and the olfactory tubercle. When multiple testing was considered using Bonferroni’s correction, significant differences survived only in the dorsolateral, dorsomedial, and ventrolateral caudate-putamen, as well as in the medial septum (*p* < 0.001).

### 2.2. Generation and Comparison of the Correlation Matrices for the Connectivity Analysis

We used *Homer1a* expression levels to calculate all pairwise correlation coefficients between pairs of ROIs for each treatment group (please see Appendix A for Pearson’s *r* and *p*-values) and generated four correlation matrices (please see Figure 2).

It is noteworthy that the administration of ASE is associated with the appearance of multiple negative correlations between ROIs in the VEH/ASE matrix. It should be noted that methods for comparing brain networks largely ignore negative correlations [17] even if negative edges may be neurobiologically relevant and their significance is yet to be clarified [18]. In this case, the inter-correlation between ROIs, including the nucleus accumbens, cingulate cortex, motor cortex, and striatal subregions became negative.

Moreover, acute KET challenge was associated with the appearance of negative correlations between the region corresponding to the medial part of the cingulate cortex and indusium griseum, and all remaining ROIs. Since the indusium griseum receives dense dopamine afferents and contains dopaminergic neurons, this region has been described as a common neuronal target of psychostimulant action [19,20]. Given the effects of KET on the dopamine function [21], it is possible that KET has similar effects to amphetamines on this specific brain region, which is classically considered as a part or a remnant of the hippocampus.

The KET/ASE matrix was characterized by a pattern of stronger and positive connections, a large portion of which are significant or highly significant, as shown in Figure 2. The connections between caudate-putamen subdivisions, as well as between insular portions appear strong and positive, similar to what also occurs in the VEH/VEH matrix.

In summary, the correlation matrix most closely resembling by visual inspection that observed under physiological conditions (i.e., VEH/VEH group was that associated with the KET/ASE treatment, in which glutamatergic dysfunctions were corrected by antipsychotic administration).

By using the permutation test, we compared the edge weight of pairs of matrices (VEH/VEH vs. VEH/ASE; VEH/VEH vs. KET/VEH; KET/VEH vs. KET/ASE). Significant differences are graphically displayed in Figure 3. For a comprehensive acknowledgement of significant and non-significant permutated *p*-values, please refer to Appendix A.

Among others, ASE administration was found to significantly impact correlations between subcortical and cortical ROIs in comparison to VEH injection. Specifically, the correlation between the ventral insular cortex and the ventrolateral (*p*-value after permutation test = 0.02) and dorsolateral (*p*-value after permutation test = 0.02) caudate-putamen were found to be reduced after ASE challenge, as well as the links between the somatosensory areas (S1FL and S1j) and the agranular ventral insular area (*p*-values after permutation test = 0.03 and 0.01, respectively), the claustrum (*p*-values after permutation test = 0.02 and 0.02, respectively), and the dorsal endopiriform nucleus (*p*-values after permutation test = 0.02 and <0.05, respectively).

KET challenge resulted in a reduction of Pearson’s *r* coefficient in multiple pairs of correlation between insular ROIs and several cortical and subcortical regions. Of interest, the KET/VEH group, when compared to VEH/VEH, exhibited a significant reduction in the correlation between the intermediate lateral septal nucleus and indusium griseum (*p*-values after permutation test = 0.01), a remnant of the former part of the hippocampus in animals.

The administration of the antipsychotic after acute KET challenge inverted Pearson’s *r* coefficient in multiple correlations between medial cingulate cortex/indusium griseum and several basal nuclei (Figure 3b), including the intermediate (*p*-values after permutation test = 0.01), dorsal (*p*-values after permutation test = 0.04), and ventral (*p*-values after permutation test = 0.04) lateral septal nuclei, the dorsolateral (*p*-values after permutation test = 0.04) and ventrolateral caudate-putamen (*p*-values after permutation test < 0.05), the upper lip of the primary somatosensory cortex (*p*-values after permutation test < 0.05), and the claustrum (*p*-values after permutation test < 0.05).

In summary, the differences observed between VEH/VEH and VEH/ASE mainly affect the correlations between cortical-subcortical regions, probably mediating the therapeutic, as well as motor side effects of antipsychotics. Moreover, the differences between VEH/VEH and KET/VEH involve interconnections starting from insular, limbic, and hippocampal ROIs. Lastly, the administration of ASE after KET challenge appears to reverse the negative intercorrelations between the medial cingulate cortex/indusium griseum and several basal nuclei.

### 2.3. Construction and Comparison of the Brain Networks

Networks were drawn as indirect graphs, with edges indicating a two-way relationship. We retained only significant correlations with a *p*-value < 0.05 in order to achieve a trade-off between sensitivity and specificity. The color of the nodes was assigned depending on the degree. To facilitate interpretation, we have positioned the network nodes on the corresponding ROIs in the Paxinos rat atlas (please see Figure 4).

Further, we calculated a series of parameters for each network, i.e., number of nodes, number of edges, network density, characteristic path length, connected components, clustering coefficient (please see Table 5), and centrality measures, including node degree and betweenness centrality (please see Appendix A).

Then, we compared the overall network properties and centrality measures, such as betweenness (please see Appendix A) and node degree (please see Appendix A), by permutation testing.

When comparing the networks in terms of global strength by permutation testing, the VEH/VEH network did not significantly differ from KET/VEH (*p*-value after permutation test = 0.547), nor KET/VEH differed from KET/ASE (*p*-value after permutation test = 0.45), whereas VEH/ASE exhibited a significantly reduced global strength compared to VEH/VEH (*p*-value after permutation test = 0.049).

With regard to the node centrality metrics of VEH/VEH vs. VEH/ASE networks, the degree was significantly different in Cl (*p*-value after permutation test = 0.013), CPDL (*p*-value after permutation test = 0.018), CPDM (*p*-value after permutation test = 0.04), CPVL (*p*-value after permutation test = 0.024), S1FL (*p*-value after permutation test = 0.02), and Den (*p*-value after permutation test = 0.02), while the betweenness differed only in Cg2 (*p*-value after permutation test = 0.017) and LSV (*p*-value after permutation test = 0.031).

By comparing VEH/VEH and KET/VEH networks, the degree was not different among nodes, while the betweenness was significantly higher in M2 (*p*-value after permutation test = 0.022) and MS (*p*-value after permutation test = 0.042) after acute KET challenge.

Finally, the betweenness of S1DZ (*p*-value after permutation test = 0.009), CPDM (*p*-value after permutation test = 0.009), LSV (*p*-value after permutation test = 0.017), Ig (*p*-value after permutation test = 0.032), Shi (*p*-value after permutation test = 0.035), Pir (*p*-value after permutation test = 0.04), and Cg1 (*p*-value after permutation test = 0.045) was significantly different between KET/VEH and KET/ASE networks, while no difference in degree was observed. In particular, the betweenness of Ig, Cg1, Shi, and LSV was lower in the KET/ASE network, while the betweenness of S1DZ, CPDM, and Pir was higher in the KET/ASE network.

Thus, the VEH/ASE group was associated with decreased betweenness of the cingulate cortex and lateral septal ROIs compared to the VEH/VEH group. The administration of KET after VEH pre-treatment was associated with a higher betweenness in the supplementary motor cortex and medial septum nodes compared to VEH/VEH. Finally, the administration of antipsychotics after the KET challenge was able to increase the betweenness of multiple nodes, while reducing the betweenness of others with respect to the KET/VEH network.

We have summarized the main findings of our study in Table 6.

## 3. Discussion

In the present experiment, we evaluated whether the expression of the IEG *Homer1a* in multiple cortical and subcortical ROIs was affected by the treatment with the second-generation antipsychotic ASE, administered alone in naïve (i.e., VEH pre-treated) rats or KET pre-treated rats. Based on previously published papers, acute KET administration is regarded as a valuable and heuristic preclinical model of psychosis [22]. It has been documented that KET treatment in humans (at dose levels comparable to those utilized in our investigation) causes behavioral and neurochemical effects mimicking psychosis, including the multifaceted symptom presentation [23,24].

We did not observe a significant induction of *Homer1a* by acute KET administration at the timing chosen for the animal sacrifice after the treatments. Following the normalization of the data on values of gcc, a region that should not deliver signal intensity, *Homer1a* expression values in the KET/VEH group were even lower than in the control group, although the significance did not survive the Bonferroni correction. These results differ from a previous report from our group, which instead found a dose-dependent increase in *Homer1a* levels after KET administration [25]. However, the inconsistency in findings could be attributable to the different animal treatment procedures due to the administration of saline after KET challenge and the interval of additional 30 min before animal sacrifice in the present experiment. Hence, it is possible that, after acute KET exposure, *Homer1a* transcript levels increased for 90 min and returned to approximately baseline values in 120 min. We may therefore have captured different moments of the *Homer1a* expression curve following the challenge with an NMDAR antagonist. In a previous work by Buonaguro et al., 2017, exploring the effects at a post-synaptic level of antipsychotics and minocycline, both in a naturalistic context and after KET challenge, the authors did not perform gene expression comparisons between groups receiving different pre-treatments, and comparisons were separately carried out between VEH pre-treated groups on the one hand, and KET pre-treated groups on the other [26].

ASE administration in VEH pre-treated animals produced a region-specific pattern, inducing *Homer1a* only to a limited extent and never reaching significance. Again, after normalization, *Homer1a* values were higher in the Cg2 and vp in the control group than in the VEH/ASE group, although significance did not survive the Bonferroni correction.

Lastly, the administration of ASE in an animal model of acute NMDAR dysfunction obtained by acute KET challenge was able to upregulate *Homer1a* almost in all of the regions considered. In particular, significant differences survived in the medial septum, dorsolateral, dorsomedial, and ventrolateral caudate-putamen when multiple testing correction was taken into account. These findings are consistent with previous reports showing that ASE only mildly impacts the cortical gene expression [27]. ASE relevant action in KET pre-treated rats, paralleled by the failure to detect an increase in *Homer1a* transcript levels in VEH pre-treated rats, may indicate that antipsychotics preferentially deliver their effects in a context of altered glutamatergic functions much more than under physiological conditions.

It has been argued that the extent of *Homer1a* induction may be secondary to the degree of dopamine receptor blockade and the specific subtype [28]. Given the peculiar synergism of D1Rs and NMDARs, ASE effects on *Homer1a* may depend on its action at D1R sites [29]. *Homer1a* induction may also be triggered by 5-hydroxytryptamine 2A receptor (5-HT2AR) antagonism, which positively affects glutamatergic transmission [30]. However, repeated ASE exposure in animal models has been associated with a decreased 5-HT2AR binding in the medial prefrontal cortex and dorsolateral frontal cortex but not in other brain regions [31]. Moreover, striatal density of D1Rs is high whereas that of 5-HT2ARs is low, thus ASE-induced striatal gene expression could be mainly driven by the action at D1R sites [32].

As it can be inferred from the correlation matrices, the topographical organization of the four functional networks (i.e., VEH/VEH, VEH/ASE, KET/VEH, KET/ASE) varied widely, especially for the link between the cortex and striatum. In the present experiment, we observed that the VEH/VEH network was characterized by stronger functional connections between AIV and lateral caudate-putamen (CPVL and CPDL) compared to the VEH/ASE network. Ventral caudate-putamen and insular regions have been involved in the assignment of emotional value and reward magnitude expectation [33]. The links between the somatosensory areas (S1FL and S1j) and AIV, Cl, and Den (the latter two ROIs belonging to the amygdala complex [34]) are reduced in the VEH/ASE network. As well known, the insula receives sensory inputs from the somatosensory cortices relevant to pain sensitivity [35]. Antipsychotic ability to target discrete insular connections with striatal and somatosensorial regions might account for their effects on perception, motivation, and salience assignment.

Acute KET challenge in the KET/VEH group is associated with negative correlations between the ROI corresponding to the medial portion of the cingulate cortex and the indusium griseum, and remaining ROIs, which are not observed in other treatment groups. It is noteworthy that indusium griseum has been described as a vestigial structure in humans and a remnant of the former part of the hippocampus in animals. Hippocampus is central in the neurobiology of psychotic disorders and the perturbation of functional connectivity within the hippocampus, as well as its extrinsic connections has been considered to contribute to schizophrenia deficits much more than psychotic symptoms [36]. However, only indusium griseum correlation with LSI (part of the septal area, the anterior portion of the limbic system) was significantly weakened in the comparison between VEH/VEH and KET/VEH.

Moreover, it should be noted that the KET/ASE group showed significantly higher inter-correlations between the medial portion of the cingulate cortex and lateral putamen, the upper lip of the primary somatosensory cortex, septal area nuclei, and claustrum, in comparison to the KET/VEH group. Since brain sections were quantitated at the topographical level of the striatum, we were unable to directly investigate the connectivity of hippocampal regions. However, indusium griseum behaves as a single functional and neuroanatomical unit together with the anterior aspect of the hippocampus [37], a candidate region in the study of schizophrenia and a functional hub for multiple brain networks [38]. Altered hippocampal-striatal coupling has been reported to be involved in deficits in associative learning tasks [38]. In this context, it is noteworthy that ASE administration in KET pre-treated rats appears to reverse Pearson’s *r* coefficient in medial cingulate cortex-caudate correlations.

Finally, we analyzed the global strength and indices of centrality in each network and identified discrete nodes with enhanced centrality metrics. Although ASE does not directly recruit these regions by inducing *Homer1a* expression, it was able to finely regulate the centrality of the cingulate cortex ROIs. The cingulate cortex is implicated in the regulation of cognitive, sensorimotor, affective, and visceral functions [39]. The centrality of several brain regions, including olfactory, medial, and superior frontal regions, anterior cingulate, medial temporal pole, and superior occipital regions has been found impaired in functional connectomic studies conducted on schizophrenia patients [40]. We may therefore conceive that ASE ability to modulate the betweenness of the cingulate cortex area 2 (corresponding to the anterior cingulate cortex) both in VEH and KET pre-treated rats may contribute to its antipsychotic action. Of interest, a magnetic resonance spectroscopy study suggested that elevations in glutamate and glutamate metabolites in the anterior cingulate cortex predicted a poor antipsychotic response [41]. Since Homer proteins are expressed limitedly at glutamatergic synapses, ASE ability to reduce the betweenness of this region in the Homer1a-based network is particularly attractive.

The administration of ASE in the KET pre-treated group was found to significantly modify the betweenness of the LSV compared to KET/VEH. Since lateral septal nuclei have been recently identified as critical hubs linking hippocampal and prefrontal activity with subcortical areas, participating in cognitive functions and motivated behaviors, this action may account for the beneficial impact on negative symptoms in psychosis [42]. It is noteworthy that ASE administration after KET challenge was associated, in our study, with a decrease in the betweenness of Ig compared to the KET/VEH network. According to Kraguljac et al., glutamate levels in hippocampal regions, as detected by magnetic resonance spectroscopy, were significantly higher in schizophrenia patients compared to controls. Nonetheless, they failed to reveal any effect of the antipsychotic treatment with risperidone on hippocampal glutamate concentrations. Since Homer1a is a marker of glutamatergic connectivity, it follows that, although ASE does not significantly affect gene expression in the Ig, it may reduce the centrality of this region in the global glutamatergic connectivity network.

We have tested the action of the antipsychotic compound on a pharmacological model of psychosis; however, further analyses of the gene expression connectome could take advantage of genetic rodent models mirroring persistent reorganization of neural circuitry characteristic of schizophrenia, such as disrupted in schizophrenia 1 (DISC1) L100P mutants [43].

In conclusion, we detected significant differences in node and edge measures across groups exposed to different treatments, highlighting ASE capability to finely regulate brain connectivity by shaping synaptic architecture and restoring a functional pattern of interregional co-activation. Further studies are warranted to disentangle the contribution of different components of the antipsychotics’ receptor profile in modifying the pattern of connectome alterations induced by NMDAR non-competitive blockade; this could be instrumental to design compounds that can better counteract the NMDAR hypofunction considered relevant for the pathophysiology of psychosis.

## 4. Materials and Methods

### 4.1. Animals

According to a previously published protocol, we conducted the ISHH procedures in Sprague–Dawley rats [44], in light of the evidence supporting the use of ketamine-treated Sprague–Dawley rats for modelling schizophrenia [22,26,45]. Sprague–Dawley rats from Charles River Labs were chosen because they are among the most commonly used rat strains, easily purchased from the above vendor. Male Sprague–Dawley rats (*n* = 20) with an average weight of 250 g (Charles-River Labs, Lecco, Wilmington, MA, USA) were housed and adapted to human handling in a temperature and humidity-controlled colony room, kept on a 12 h/12 h light/dark cycle with ad libitum access to food and water. The experimental procedures and animal handling techniques were conducted in agreement with the NIH guide for care and use of laboratory animals (NIH publication no. 85–23, revised 1996) and approved by local animal care and use committee. All efforts have been made to minimize animal suffering.

### 4.2. Drug Treatment and Study Design

To assess gene expression both under vehicle treatment and after manipulation of the glutamatergic system, animals were randomly assigned to two groups (*n* = 10 for each pre-treatment group), receiving saline (VEH; NaCl 0.9%) or KET (30 mg/kg), respectively. Acute administration of KET at sub-anesthetic and sub-convulsant doses was chosen among validated preclinical models to reproduce the schizophrenia-like behavioral and neurochemical phenotype of schizophrenia [22].

Subsequently, each pre-treatment group was randomly split into two arms, receiving a second compound: the atypical antipsychotic ASE (0.3 mg/kg), or saline (VEH) (*n* = 5 for each treatment group). The second compound was administered intraperitoneally (i.p.) 30 min after pre-treatment. Since the Homer1a peak expression occurs 90–120 min after psychotropic drug challenges [46,47,48], we opted for a 30-min interval between injections in order to capture the Homer1a expression resulting from the combination of the two compounds, prior to the effect of the first compound no longer being detectable.

ASE was administered at behaviorally active doses, known to induce gene expression according to previous published experimental protocols [49,50]. Thus, the following four treatment groups were obtained: (a) VEH + VEH, (b) VEH + ASE, (c) KET + ASE, and (d) KET + VEH. Animals were sacrificed by decapitation 90 min after the second injection. KET (Sigma-Aldrich, St. Louis, MO, USA) and ASE (Lundbeck A/S, Copenhagen, Denmark) were supplied as a powder and dissolved in saline solution (NaCl 0.9%), adjusted to physiological pH, and injected i.p. at the final volume of 1 mL/kg.

### 4.3. In Situ Hybridization Histochemistry (ISHH)

Brains were quickly removed and frozen on dry powdered ice, and then stored at −70 °C until sectioning. Coronal brain slices (12 μm) were cut on a cryostat using the Paxinos rat atlas as a reference [51]. Sections were thaw-mounted onto gelatin-coated slides and stored at −70 °C for further analysis. The probes used for radioactive ISHH were oligodeoxyribonucleotide with a length of 48 bp, complementary to the mRNA sequence (bases 2527–2574) of the *Homer1a* gene (GenBank #U92079; MWG Biotech, Firenze, Italy). The probe sequence was derived from those used in our previous hybridization studies investigating the *Homer1a* expression [52]. The probes were labeled using ^35^S as a radioisotope and sections were processed for radioactive ISHH according to previously published protocols [44]. Hybridized sections were dried and exposed to Kodak-Biomax MR Autoradiographic film (Sigma-Aldrich, Milano, Italy). A slide containing a scale of 16 known amounts of ^14^C standards (ARC-146C, American Radiolabeled Chemical, Inc., St. Louis, MO, USA) was co-exposed with the samples. Each slide contained three adjacent brain sections of a single animal. The autoradiographic films were exposed in a time range of 10–45 days. The optimal time of exposure was chosen to maximize the signal-to-noise ratio but to prevent optical density from approaching the limits of saturation. Each slide contained three adjacent brain sections from a single animal. The autoradiographic films were exposed at a time interval between 10 and 45 days. The optimal exposure time was chosen to maximize the signal-to-noise ratio but to avoid the optical density approaching saturation limits.

### 4.4. Image Analysis

The autoradiographic films have been digitized by a transparency film scanner (Microtek 9800XL Plus TMA) preserving their original characteristics. Then, the optical density of the autoradiographic signal was quantitated by using ImageJ software (v. 1.46v, http://rsb.info.nih.gov/ij/ [accessed on 5 December 2022]) in 33 different ROIs at the topographical level of the striatum (from Bregma +1.68 to +1.44). The optical density evaluation was carried out by two independent investigators. For each animal, values from three adjacent sections were averaged and the mean value was reported with standard deviation in relative dpm.

### 4.5. Statistical Analysis

By averaging the measurements from three adjacent sections of each animal brain, we calculated the gene expression values in each region. Further, data were normalized by values of the *Homer1a* gene expression in the gcc, which should not deliver signal intensity. The Shapiro–Wilk test was used to determine if the relative dpm values were distributed normally. Repeated measures analysis of variance (ANOVA) was used to determine the individual contribution of each of the categorization factors on the outcome of the dependent variable (i.e., *Homer1a* gene expression). We analyzed the effect of the between-subjects variable (treatment) as well as the within-subject variable (ROI) effects and their interaction. Moreover, we used Student’s *t* test to compare the transcript values of *Homer1a* in (i) VEH/VEH vs. KET/VEH groups, in order to evaluate the *Homer1a* expression in the presence or absence of a challenge of the glutamatergic system; (ii) VEH/VEH vs. VEH/ASE groups, in order to understand the effect of the antipsychotic on gene expression in baseline conditions; (iii) KET/VEH vs. KET/ASE groups, to assess the effects of ASE on the *Homer1a* transcript levels in an animal model of schizophrenia. The comparisons were performed by using Student’s *t*-test. The threshold for comparisons’ significance was set at 0.05. Multiple testing error was managed by using the Bonferroni correction (adjusted *p*-value = 0.0015). For statistical analysis, SAS Institute Inc.’s JMP software version 9.0.1 and IBM SPSS 25 were adopted.

#### Comparison of the Correlation Matrices and Generation of Networks

By using the *Homer1a* signal intensity measures in each ROI as dependent variables, we calculated Pearson’s *r* for all possible pairwise correlations in each treatment group (i.e., VEH/VEH, VEH/ASE, KET/VEH, and KET/ASE) and four correlation matrices were generated. The statistical analyses and graphical outputs were obtained via the software R.4.2.1 with the “hmisc”, “corrplot”, and “dnt (differential network tests)” packages (http://www.r-project.org/ [accessed on 19 December 2022]), as well as Cytoscape software 3.8.2 (http://www.cytoscape.org/ [accessed on 19 December 2022]). We calculated the network properties, such as the characteristic path length, clustering coefficients, network density, and connected components. We used a function providing a permutation-based test for comparing networks and calculating significant differences between paired edges [53]. Further, we assessed the differences between networks in the global strength and other basic centrality properties, such as nodes’ degree and betweenness centrality through a permutation-based approach [53]. The significance threshold was set at a *p*-value of 0.05 as used in previous studies [53]. A large number of permutations (*n* = 1000) was employed to obtain reliable results [53].

Networks consisting of 33 nodes (as many as the ROIs investigated) were graphically generated; each network was summarized by its weighted adjacency matrix, where the edge weights between the two nodes refer to the corresponding *r* correlation coefficient value, ranging from −1 to +1, indicating the magnitude or strength of an edge. Graphical outputs were obtained by styling the edges based on their weights, and nodes based on the degree. In an effort to retain only relevant edges and avoid spurious ones, we filtered significant correlations with a minimum *p*-value < 0.05 to achieve a trade-off between sensitivity and specificity.

The method applied in the present article, including the study protocol, ISHH procedures, the computation of correlation matrices, and network elaboration are summarized in the graphical abstract.

## Figures and Tables

**Figure 1 ijms-24-04372-f001:**
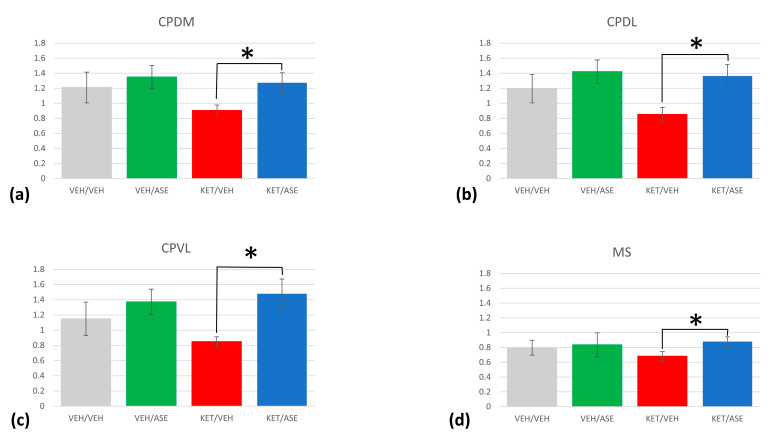
*Homer1a* mRNA levels in different treatment groups expressed as relative dpm ± sem. Significant differences between KET/VEH and KET/ASE groups were found in the dorsomedial (**a**), dorsolateral (**b**), ventrolateral (**c**) caudate-putamen, as well as in the medial septum (**d**). Asterisks denote statistically significant differences (Bonferroni adjusted *p*-value ≤ 0.001). CPDM =dorsomedial caudate-putamen; CPDL = dorsolateral caudate-putamen; CPVL = ventrolateral caudate-putamen; MS = medial septum.

**Figure 2 ijms-24-04372-f002:**
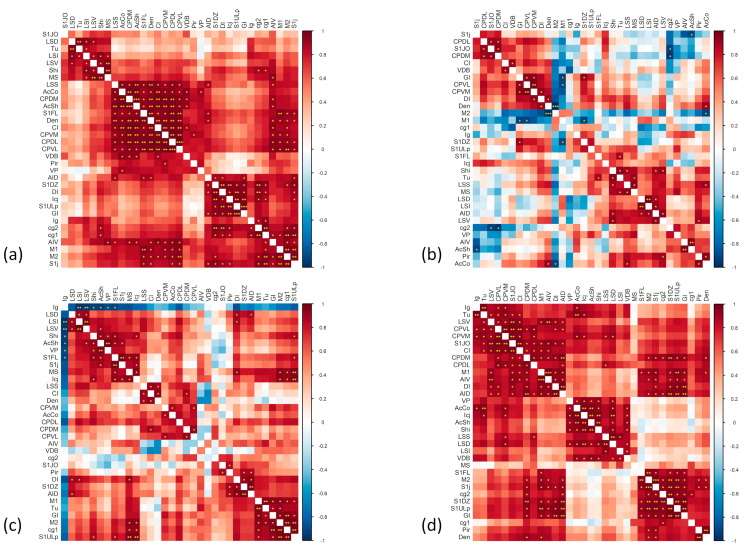
Correlation matrix for VEH/VEH (**a**), VEH/ASE (**b**), KET/VEH (**c**), and KET/ASE (**d**) group. Variables are grouped according to hierarchical clustering. According to the color scale on the right of the matrix, positive correlations are reported in red, negative ones in blue. Asterisks represent significance levels, accordingly: *p*-values < 0.05 are indicated with an asterisk, *p*-values < 0.01 with two asterisks, *p*-values < 0.001 with three asterisks. VEH: vehicle; KET: ketamine; ASE: asenapine.

**Figure 3 ijms-24-04372-f003:**
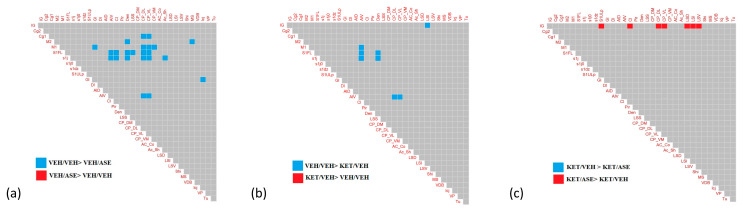
Differences in the pattern of correlation coefficients between groups are displayed in panel (**a**) (VEH/VEH vs. VEH/ASE); panel (**b**) (VEH/VEH vs. KET/VEH); and panel (**c**) (KET/VEH vs. KET/ASE). Blue squares indicate pairs of correlations in which significant differences are positive, red squares indicate couples of correlations in which significant differences are negative. VEH = vehicle; ASE = asenapine; KET = ketamine.

**Figure 4 ijms-24-04372-f004:**
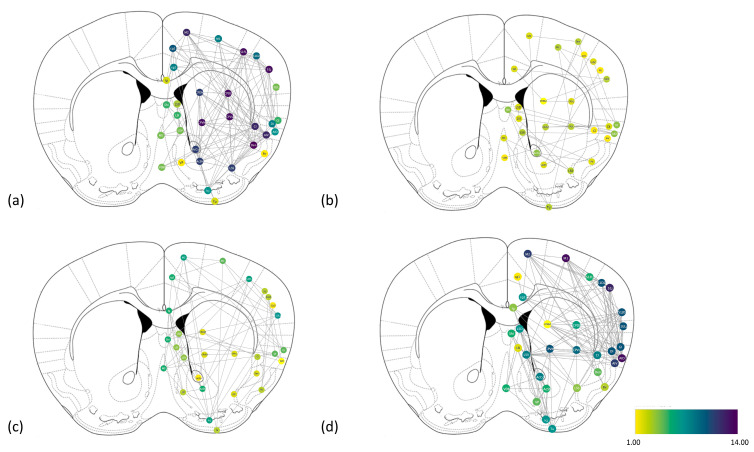
Graph representation of the brain networks for VEH/VEH (**a**), VEH/ASE (**b**), KET/VEH (**c**), and KET/ASE (**d**) groups. Nodes represent ROIs and the color reflects the corresponding degree.

**Table 1 ijms-24-04372-t001:** Regions of interest (ROIs) we quantitated in the present study and corresponding abbreviations.

Abbreviation	Brain Region
LSV	Lateral septal nucleus, ventral
Tu	Olfactory tubercle
LSD	Lateral septal nucleus, dorsal
LSI	Lateral septal nucleus, intermediate
Icj	Islands of Calleja
VP	Ventral pallidum
Shi	Septohippocampal nucleus
MS	Medial septum
VDB	Nucleus of the vertical limb of the diagonal band
IG	Indusium griseum
S1dz	Somatosensory 1, dysgranular zone
AIV	Agranular insular area, ventral
S1ULp	Upper lip of the primary somatosensory cortex
GI	Granular insular cortex
DI	Dysgranular insular cortex
AcSh	Accumbens nucleus, shell
Pir	Piriform cortex
AID	Agranular insular area, dorsal
LSS	Lateral stripe of striatum
Cg2	Cingulate cortex, area 2
CPDL	Dorsolateral caudate putamen
CPVL	Ventrolateral caudate putamen
CPVM	Ventromedial caudate putamen
Cg1	Cingulate cortex, area 1
M2	Supplementary motor cortex
CPDM	Dorsomedial caudate putamen
Den	Dorsal endopiriform nucleus
M1	Primary motor cortex
S1Fl	Somatosensory 1, forelimb region
AcCo	Accumbens nucleus, core
Cl	Claustrum
S1j	Somatosensory 1, jaw region
S1jO	Primary somatosensory cortex, jaw region, oral surface

**Table 2 ijms-24-04372-t002:** Results of Student’s *t* test comparing the VEH/VEH vs. KET/VEH groups throughout the 33 regions of interest. Significant values were outlined in bold. ROIs: regions of interest.

ROIs	*t*	Degree of Freedom	*p*-Value	Confidence Interval	
				Inferior	Superior
Ig	1.29	8	0.234	−0.31	0.11
Cg2	−3.48	8	**0.008**	−0.47	−0.96
Cg1	−4.43	8	**0.002**	−0.64	−0.20
M2	−3.56	8	**0.007**	−0.59	−0.13
M1	−3.47	8	**0.008**	−0.51	−0.10
S1FL	−3.22	8	**0.012**	−0.44	−0.73
S1j	−2.59	8	**0.032**	−0.38	−0.02
S1jO	0.05	8	0.961	−0.76	0.79
S1DZ	−2.28	8	0.052	−0.37	0.00
S1ULp	−1.22	8	0.259	−0.27	0.08
GI	−1.03	8	0.335	−0.35	0.14
DI	−0.70	8	0.506	−0.26	0.14
AID	−0.45	8	0.662	−0.28	0.19
AIV	−0.16	8	0.880	−0.15	0.13
Cl	−0.97	8	0.363	−0.30	0.12
Pir	−2.07	8	0.073	−0.44	0.02
Den	−1.13	8	0.292	−0.35	0.12
LSS	−1.48	8	0.176	−0.32	0.07
CPDM	−3.14	8	**0.014**	−0.53	−0.08
CPDL	−3.63	8	**0.007**	−0.56	−0.12
CPVL	−2.93	8	**0.019**	−0.53	−0.64
CPVM	−4.09	8	**0.003**	−0.66	−0.18
AcCo	−2.74	8	**0.026**	−0.48	−0.04
AcSh	−2.45	8	**0.040**	−0.56	−0.02
LSD	−1.78	8	0.113	−0.29	0.04
LSI	−1.47	8	0.180	−0.26	0.06
LSV	−2.58	8	**0.033**	−0.27	−0.02
Shi	−2.19	8	0.060	−0.26	0.01
MS	−2.14	8	0.065	−0.24	0.01
VDB	−1.311	8	0.226	−0.29	0.08
Icj	−3.55	8	**0.007**	−0.24	−0.05
VP	−4.21	8	**0.003**	−0.35	−0.10
Tu	−2.68	8	**0.028**	−0.36	−0.03

**Table 3 ijms-24-04372-t003:** Results of Student’s *t* test comparing the VEH/VEH vs. VEH/ASE groups throughout the 33 regions of interest. Significant values were outlined in bold. ROIs: regions of interest.

ROIs	*t*	Degree of Freedom	*p*-Value	Confidence Interval	
				Inferior	Superior
Ig	0.92	8	0.387	−0.05	0.12
Cg2	−2.43	8	**0.041**	−0.36	−0.01
Cg1	−2.16	8	0.063	−0.45	0.02
M2	−1.92	8	0.091	−0.40	0.04
M1	−1.69	8	0.130	−0.38	0.06
S1FL	−1.53	8	0.164	−0.32	0.07
S1j	−1.28	8	0.235	−0.30	0.09
S1jO	0.42	8	0.687	−0.76	1.09
S1DZ	−0.54	8	0.607	−0.21	0.13
S1ULp	−0.10	8	0.925	−0.18	0.17
GI	−0.43	8	0.677	−0.31	0.21
DI	−0.14	8	0.889	−0.23	0.20
AID	−0.42	8	0.686	−0.26	0.18
AIV	−0.88	8	0.403	−0.20	0.09
Cl	1.69	8	0.129	−0.06	0.39
Pir	−1.77	8	0.114	−0.38	0.05
Den	1.22	8	0.258	−0.15	0.50
LSS	0.92	8	0.386	−0.14	0.32
CPDM	1.22	8	0.256	−0.12	0.40
CPDL	2.07	8	0.073	−0.03	0.48
CPVL	1.81	8	0.108	−0.06	0.51
CPVM	0.17	8	0.867	−0.25	0.29
AcCo	0.13	8	0.901	−0.24	0.27
AcSh	−0.12	8	0.912	−0.28	0.25
LSD	−0.17	8	0.867	−0.17	0.15
LSI	−0.04	8	0.968	−0.15	0.15
LSV	−0.23	8	0.825	−0.11	0.09
Shi	0.33	8	0.749	−0.16	0.21
MS	0.46	8	0.657	−0.16	0.24
VDB	0.75	8	0.474	−0.12	0.24
Icj	−0.44	8	0.669	−0.17	0.12
VP	−2.67	8	**0.028**	−0.23	−0.02
Tu	−1.68	8	0.131	−0.35	0.05

**Table 4 ijms-24-04372-t004:** Results of Student’s *t* test comparing the KET/VEH vs. KET/ASE groups throughout the 33 regions of interest. Significant values were outlined in bold. Significant adjusted *p*-values after the Bonferroni correction were given in red. ROIs: regions of interest.

ROIs	*t*	Degree of Freedom	*p*-Value	Confidence Interval	
				Inferior	Superior
Ig	−1.17	8	0.275	−0.16	0.05
Cg2	−3.00	8	**0.017**	−0.40	−0.05
Cg1	−2.59	8	**0.032**	−0.49	−0.03
M2	−2.89	8	**0.020**	−0.52	−0.06
M1	−2.60	8	**0.032**	−0.51	−0.03
S1FL	−3.19	8	**0.013**	−0.55	−0.09
S1j	−2.32	8	**0.049**	−0.53	0.00
S1jO	−1.73	8	0.121	−0.69	0.10
S1DZ	−2.50	8	**0.037**	−0.53	−0.02
S1ULp	−2.30	8	0.051	−0.55	0.00
GI	−2.52	8	**0.036**	−0.61	−0.03
DI	−3.21	8	**0.012**	−0.66	−0.11
AID	−3.42	8	**0.009**	−0.63	−0.12
AIV	−2.83	8	**0.022**	−0.54	−0.05
Cl	−2.94	8	**0.019**	−0.55	−0.07
Pir	−2.66	8	**0.029**	−0.50	−0.03
Den	−2.91	8	**0.02**	−0.36	−0.04
LSS	−3.28	8	**0.011**	−0.58	−0.10
CPDM	−5.41	8	** 0.001 **	−0.52	−0.21
CPDL	−6.24	8	** <0.001 **	−0.7	−0.32
CPVL	−6.93	8	** <0.001 **	−0.84	−0.42
CPVM	−4.32	8	**0.003**	−0.71	−0.21
AcCo	−3.36	8	**0.01**	−0.58	−0.11
AcSh	−3.49	8	**0.008**	−0.63	−0.13
LSD	−3.13	8	**0.014**	−0.46	−0.07
LSI	−2.90	8	**0.02**	−0.26	−0.03
LSV	−3.14	8	**0.014**	−0.38	−0.06
Shi	−3.58	8	**0.007**	−0.36	−0.08
MS	−4.84	8	** 0.001 **	−0.29	−0.1
VDB	−2.61	8	0.06	−0.37	0.01
Icj	−2.71	8	**0.027**	−0.47	−0.04
VP	−2.70	8	**0.027**	−0.48	−0.04
Tu	−1.95	8	0.086	−0.44	0.04

**Table 5 ijms-24-04372-t005:** Discrete parameters are reported for each network after retaining only significant (*p*-value < 0.05) correlations.

Parameters	VEH/VEH	VEH/ASE	KET/VEH	KET/ASE
Number of nodes	32	28	29	32
Number of edges	128	34	60	135
Network density	0.129	0.045	0.074	0.136
Characteristic path length	1.866	1.256	1.589	2.445
Connected components	1	4	2	1
Clustering coefficient	0.328	0.193	0.255	0.332

**Table 6 ijms-24-04372-t006:** Summary of the significant findings. Asterisks represent couples of correlations in which significant differences are negative.

*Homer1a* Gene Expression Comparison			
Treatment Group Differences	ROIs		*p*-value
KET/ASE > KET/VEH	CPDM		≤0.001
KET/ASE > KET/VEH	CPDL		≤0.001
KET/ASE > KET/VEH	CPVL		≤0.001
KET/ASE > KET/VEH	MS		≤0.001
Comparison of Correlation Matrices			
Matrix pairs	ROIs		*p*-Value
VEH/VEH vs. VEH/ASE	M1/GI		≤0.05
	S1j/AIV		≤0.05
	S1FL/CPDL		≤0.05
	S1j/CPDL		≤0.05
	S1FL/CPVL		≤0.05
	S1j/CPVL		≤0.05
	S1FL/CI		≤0.05
	S1j/CI		≤0.05
	M2/Den		≤0.05
	S1FL/Den		≤0.05
	AIV/CPDL		≤0.05
	AIV/CPVL		≤0.05
	M1/CPVM		≤0.05
	S1j/AcSh		≤0.05
	S1FL/AIV		≤0.05
	S1FL/LSS		≤0.05
	M1/CPVL		≤0.05
	GI/Icj		≤0.05
	S1j/Den		≤0.05
	Cg1/CPDL		≤0.05
	M1/CPDL		≤0.05
	Cg1/CPVL		≤0.05
	M2/MS		≤0.05
VEH/VEH vs. KET/VEH	M1/AIV		≤0.05
	S1FL/AIV		≤0.05
	AIV/CPDL		≤0.05
	Ig/LSI		≤0.05
	S1j/AIV		≤0.05
	S1FL/Den		≤0.05
	S1j/Den		≤0.05
	AIV/CPVL		≤0.05
KET/VEH vs. KET/ASE	Ig/LSI *		≤0.05
	Ig/CPDL *		≤0.05
	Ig/LSD *		≤0.05
	Ig/LSV *		≤0.05
	Ig/S1ULp *		≤0.05
	Ig/CI *		≤0.05
	Ig/CPVL *		≤0.05
Comparison of Network Measures			
Matrix Pairs	Network Measures		*p*-Value
VEH/ASE vs. VEH/VEH	Global Strength		≤0.05
VEH/VEH vs. VEH/ASE	Node centrality	Cl	≤0.05
		CPDL	≤0.05
		CPDM	≤0.05
		CPVL	≤0.05
		S1FL	≤0.05
		Den	≤0.05
	Betweenness	Cg2	≤0.05
		LSV	≤0.05
VEH/VEH vs. KET/VEH	Betweenness	M2	≤0.05
		MS	≤0.05
KET/VEH vs. KET/ASE	Betweenness	S1DZ	≤0.01
		CPDM	≤0.01
		LSV	≤0.05
		Ig	≤0.05
		Shi	≤0.05
		Pir	≤0.05
		Cg1	≤0.05

## Data Availability

All data are available upon request.

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
