# Peer review of "A Postsynaptic Density Immediate Early Gene-Based Connectome Analysis of Acute NMDAR Blockade and Reversal Effect of Antipsychotic Administration"

_ijms, 2023, doi:10.3390/ijms24054372_

Round 1
Reviewer 1 Report
Overall, the material presented in the article is pertinent and invaluable for molecular biology and psychiatry. The contribution of alterations in the glutamatergic system and NMDA receptor hypofunction to the pathogenesis of schizophrenia is now actively discussed. The authors' work shows well how antipsychotic therapy can correct molecular dysfunction. Thanks to the correct processing of the data, the authors were able to obtain interesting and important results.
The authors were able to show that modulation of synaptic transmission occurs under the influence of asenapine. However, this was not always significantly pronounced. I would like to suggest that the authors confirm their results on an animal model of schizophrenia, such as DISC1-L100P. Perhaps with persistent molecular dysfunction the results would be more pronounced.
There are several observations, which, however, do not require substantial reworking of the text.
1) Figure 1 needs to be modified. Leave only the graphs with significant differences and describe the rest in the text or in the supplementary material.
2) The text lacks a validation for the selection of the Sprague-Dawley rat strain for the experiment You need to supplement the Introduction or Materials and Methods with data about why specimens of this strain are suitable for your inquiry.
3) There is no endorsement for opting for the timing of ASN administration 30 minutes after VEN and KET pretreatment and decapitation 90 minutes after ASN administration. In the discussion section, you cite a link to a journal with a dose-dependent effect of KET on Homer1a expression (Line: 279-281), but that manuscript also lacks a rationale for the time intervals. Either cite the primary source or state that you experimentally decided upon the conditions yourself. It will be simpler for readers if there is a validation or reference to works that detail these time spans in the text.
4) In the first paragraph of the introduction (Line 39), you cite two of your three citations. There are two more citations of your own in the text that are not indispensable for understanding the manuscript. Below is a listing of references that should be removed or substituted by other works.
Line 39: References 2 and 3. Delete or Replace
Line 271: References 22. Delete or Replace
Line 274: References 23. Replace
Reviewer 2 Report
The manuscript of Barone et al. entitled “A postsynaptic density Immediate early gene-based connecome analysis of acute NMDAR blockade and reversal effect of antipsychotic administration” sheds some more light on the mechanism of action of asenapine, an atypic antipsychotic drug with antagonistic action towards D2 and D1 receptors. The authors based on two observations:1. Schizophrenia dysregulates protein levels of several molecular components of dendritic, including examined Homer 1a, and 2. Antipsychotic drugs brain plasticity, metaplasticity and postsynaptic sites. This study is largely continuation of previous work mapping the haloperidol-evoked expression of Homer1a neuronal protein.
The work provides a wealth of data, as gene expression was studied in 33 regions within the cortex, the caudate-putamen, and the nucleus accumbens of rats.
My major comments are as follows:
1. There are no clear conclusions in the Abstract, e.g. the statement whether asenapine reverses changes induced by ketamine.
2. In the introduction, I lack information about neurobiological function of the chosen brain structures (in general, the cortex, the caudate putamen and the nucleus accumbens) and the relations among them in physiology and in the course of psychotic diseases.
3. Results. Since they are quite complicated, it would be desirable to sum up each subsection of Results.
4. Also, It would be worth considering to create one more table, even without numbers, displaying in a shell all data.
5. Discussion. I would like the authors to relate obtained to practical significance of ketamine- and asenamine-induced changes in function of the chosen brain structures and their inter-correlations for the course of schizophrenia and schizophrenia treated with asenapine.
Minor comments:
1. Fig. 1 is blurred even when enlarged and, therefore, difficult to analyze.
2. The same concerns Fig. 5, i.e. symbols in circles are not visible even at very high magnification.
